

# The case for extended lifespan in cooperatively breeding mammals: a re-appraisal

Jack Thorley

Large Animal Research Group, Department of Zoology, University of Cambridge, Cambridge, UK

## ABSTRACT

Recent comparative studies have suggested that cooperative breeding is associated with increases in maximum lifespan among mammals, replicating a pattern also seen in birds and insects. In this study, we re-examine the case for increased lifespan in mammalian cooperative breeders by analysing a large dataset of maximum longevity records. We did not find any consistent, strong evidence that cooperative breeders have longer lifespans than other mammals after having controlled for variation in body mass, mode of life and data quality. The only possible exception to this general trend is found in the African mole-rats (the Bathyergid family), where all members are relatively long-lived, but where the social, cooperatively breeding species appear to be much longer-lived than the solitary species. However, solitary mole-rat species have rarely been kept in captivity or followed longitudinally in the wild and so it seems likely that their maximum lifespan has been underestimated when compared to the highly researched social species. Although few subterranean mammals have received much attention in a captive or wild setting, current data instead supports a causal role of subterranean living on lifespan extension in mammals.

## INTRODUCTION

Comparative studies of birds and insects have shown that species with cooperative and eusocial breeding systems have extended lifespans (*Arnold & Owens, 1998*; *Downing, Cornwallis & Griffin, 2015*; *Keller & Genoud, 1997*; *Keller, 1998*; *Beauchamp, 2014*).
A common explanation for the association between these breeding systems and increased lifespan is that group-living reduces the extrinsic mortality of breeding individuals, which selects for greater longevity (*Lucas & Keller, 2019*). This is thought to be the case in eusocial insect societies where the 'queens' of many ant, bee and termite species live in a sheltered nest that is defended against predators by a large workforce (*Carey, 2001*; *Hölldobler & Wilson, 1990*). An alternative possibility is that high annual survival increases local breeding competition, leading to delayed dispersal, family-living and helping (*Brown, 1987*; *Griesser et al., 2017*), such that having a relatively long lifespan makes cooperative breeding more likely to evolve, and phylogenetic reconstructions of cooperative breeding supports this argument for birds (*Downing, Cornwallis & Griffin, 2015*).

Corresponding author
Jack Thorley, jbt27@cam.ac.uk

Observations from certain mammals are also suggestive of a relationship between cooperative breeding and lifespan. The social mole-rats (family Bathyergidae), for example, include some of the longest-lived mammals for their size, with the naked mole-rat *Heterocephalus glaber*, a 40 g species from the Horn of Africa providing the most extreme case. In this species, breeding females can live for more than three decades in captivity and show no apparent age-related changes in physiology or mortality rate (*Buffenstein, 2005*; *Ruby, Smith & Buffenstein, 2018*; *Ruby, Smith & Buffenstein, 2019*; though *Dammann et al. (2019)* have questioned whether the absence of Gompertzian mortality detected by *Ruby, Smith & Buffenstein (2018)* could have arisen from the structure of their data set, rather than representing a true biological effect), prompting much interest in naked mole-rats as a model organism for gerontological research. However, as well as being long-lived, mole-rats are completely subterranean and are thought to seldom be exposed to predators in their burrow systems (*Dammann & Burda, 2007*; *Sándor, 2017*), raising the question of whether it is their subterranean lifestyle, their cooperative breeding, or both that contribute to their longevity.

Two previous comparative analyses sought to separate the role of sociality and subterranean living on maximum lifespan in mammals, the first by *Williams & Shattuck (2015)*, and the second by *Healy (2015)*. These studies followed on from an earlier comprehensive analysis of lifespan variation carried out by *Healy et al. (2014)* which tested whether species that possess traits thought to reduce extrinsic mortality risk are longer lived. This initial analysis identified fossoriality (defined as 'species living in permanent burrows') as one of several traits associated with increased maximum lifespan, with fossorial species living longer than semi-fossorial and non-fossorial species after correcting for phylogeny and body size. Yet despite being performed across a large number of non-flying animals, only five species in this first analysis were categorised as fossorial, including the long-lived naked mole-rat. In response to this one specific result, *Williams & Shattuck (2015)* noted that although it is logically consistent for underground living to be associated with increased lifespan, the sociality of the mole-rats could also account for their prolonged lifespan and obscure the interpretation of any fossoriality effect if both traits were not assessed concurrently.

To test their hypothesis, *Williams & Shattuck (2015)* collated a new dataset of maximum lifespan records from ground-living mammals and defined a larger number of species on the basis of their fossoriality (either terrestrial or fossorial) and their sociality ('eusocial' or not). While Williams and Shattuck demarcated sociality on the basis of 'eusocial tendencies', we would rather refer to such species as cooperative breeders to avoid the semantic difficulties that arise when applying a definition of eusociality to a wide range of taxa (*Boomsma & Gawne, 2018*; *Burda et al., 2000*). Cooperative breeding can be more accurately defined as those species where a proportion of females do not breed regularly and have been shown to perform alloparental care in the form of direct or indirect provisioning of offspring (as per *Solomon & French, 1997*; *Lukas & Clutton-Brock, 2012a*). Analyses of phylogenetically independent contrasts on the new dataset suggested that both fossoriality and sociality (cooperative breeding) were associated with increases in maximum lifespan, but by the authors' own admission, more nuanced phylogenetic
approaches could provide a more robust assessment of these patterns. Following this recommendation, *Healy (2015)* re-analysed the dataset of *Williams & Shattuck (2015)* using a Bayesian approach that accounts for phylogenetic uncertainty by combining information from models fitted across multiple mammalian trees, and doing so, found that that sociality, and not fossoriality, drove increases in maximum lifespan in mammals.

In this study, we re-visit the question of whether sociality, fossoriality, or both, are related to increases in maximum lifespan. The intention is not to detract from the results of the previous studies, but to place these results in context, as the ongoing reference to a link between sociality and increased lifespan in mammals (*Downing, Cornwallis & Griffin, 2015*; *Lucas & Keller, 2019*) could give the perception of a robust and resolved trend in spite of an equivocal evidence base. Indeed, not all published studies have demonstrated a positive effect of cooperative breeding on maximum lifespan. *Lukas & Clutton-Brock (2012a)* compared cooperatively breeding mammals to socially monogamous mammals and found no clear difference in maximum lifespan between species with these two mating systems. As cooperative breeding is sparsely represented across the mammalian clade (less than 1% of mammals, *Lukas & Clutton-Brock, 2012b*), slight differences in data availability or analytical method, or contrasts in definitions could contribute to inconsistencies in results across studies, and it is therefore worthwhile reflecting on how and why different studies might come to reach different conclusions. To this end, we build on previous work by using a more stringent definition of sociality along with a definition of fossoriality more aligned with the original definition in *Healy et al. (2014)* to test the role of cooperative breeding and fossoriality in driving increases in maximum lifespan. Although the use of maximum lifespan records as a proxy for lifespan is not without criticism (*Baylis, De Lisle & Hauber, 2014*), the absence of detailed life table information for many species usually precludes the use of alternative ageing metrics in a comparative setting. The use of lifespan information from captive animals could also be criticised. Clearly, captivity provides a safe environment where individuals are buffered from environmental stochasticity, predation and resource competition, and veterinary care largely eliminates disease and morbidity (*Tidière et al., 2016*). Studies of senescence in captive populations therefore make the assumption that demographic features of senescence in captivity, including maximum lifespan, are a product of (or are proportional to) selection on mortality in the wild and therefore provide useful information on the intrinsic deterioration of a species (*Carey, 2003*; *De Magalhães, Costa & Church, 2007*).

Our approach expands upon the two previous treatments of this topic in several important respects. The first major difference is in the categorisation of species according to their use of the underground environment. Here, we separate species according to whether they are subterranean feeders or not. Subterranean feeders spend almost their entire lives underground and are therefore likely to experience the low levels of predation and extrinsic mortality that should select for longer lifespans (*Hartman, 1995*; *Novikov & Burda, 2013*). In contrast, *Williams & Shattuck (2015)* chose a more relaxed definition, subsequently followed by *Healy (2015)*, whereby all mammals that make extensive use

of the underground environment were classed as fossorial. This definition forces one to group subterranean feeders (such as the moles, mole-rats, pocket gophers, or coruro *Spalacopus cyanus*) with species that use the underground environment extensively for resting, denning, and sometimes food storage, but which otherwise spend a large part of their daily activity period foraging above-ground (for example armadillos, aardvark *Orcyteropus afer*, or Eurasian badger *Meles meles*). It is not clear that mammals falling into this latter category should experience reduced predation rates to the same extent as subterranean feeders (*Healy et al., 2014* for similar interpretation), and grouping the two classes could therefore prevent the detection of an effect of subterranean living on lifespan. Secondly, although previous studies tried to overcome possible issues of sample size by using sensitivity analyses—either by removing outliers with small sample sizes (*Williams & Shattuck, 2015*), or by comparing the results of analyses where species with low sample size were included or excluded (*Healy et al., 2014*)—they did not explicitly include sample size in their modelling framework. It is known that a large number of lifespan records are needed before a maximum lifespan estimate begins to asymptote (*Moorad et al., 2012*), and since cooperative breeders are more frequently studied and kept in captivity, then all else being equal, the maximum lifespans of cooperative breeders are anticipated to be greater than non-cooperatively breeding species for purely numerical reasons (see 'Methods'); so sample size needs to be controlled for and estimated alongside the effect of cooperative breeding (*Kamilar, Bribiescas & Bradley, 2010*; *Minias & Podlaszczuk, 2017*). Thirdly, by focussing on ground-dwelling (i.e. non-arboreal) mammals, previous studies have not considered lifespan data from the callitrichid primates (the marmosets and tamarins), which represent a sizeable fraction of the total pool of cooperatively breeding species. Here, we include information from callitrichid primates.

Our study carried out an analysis of maximum lifespan across 719 mammals, controlling for sociality, subterranean living, mode of life, data quality, and body mass. To account for phylogenetic uncertainty the analysis was performed in a Bayesian framework that combined information from multiple mammalian trees. We then repeated the analysis on different subsets of the global dataset to investigate the possible influence of outlier species and data structure on estimated effect sizes.

## MATERIALS AND METHODS

### Global dataset

Maximum lifespan data was taken from the AnAGE database for 719 terrestrial mammal species (Fig. 1; *De Magalhães & Costa, 2009*). This dataset provides estimates of the sample size for each longevity record, reflecting orders of magnitude in the number of specimens that contributed to the record. Species with fewer than 10 specimens contributing to their record ('tiny') were excluded. The only exception to this data restriction was the Cape mole-rat *Georychus capensis*, which was retained because of the general lack of longevity data for African mole-rats, family Bathyergidae; for model fitting purposes this species was reclassified as having 'small' sample size. This left 323

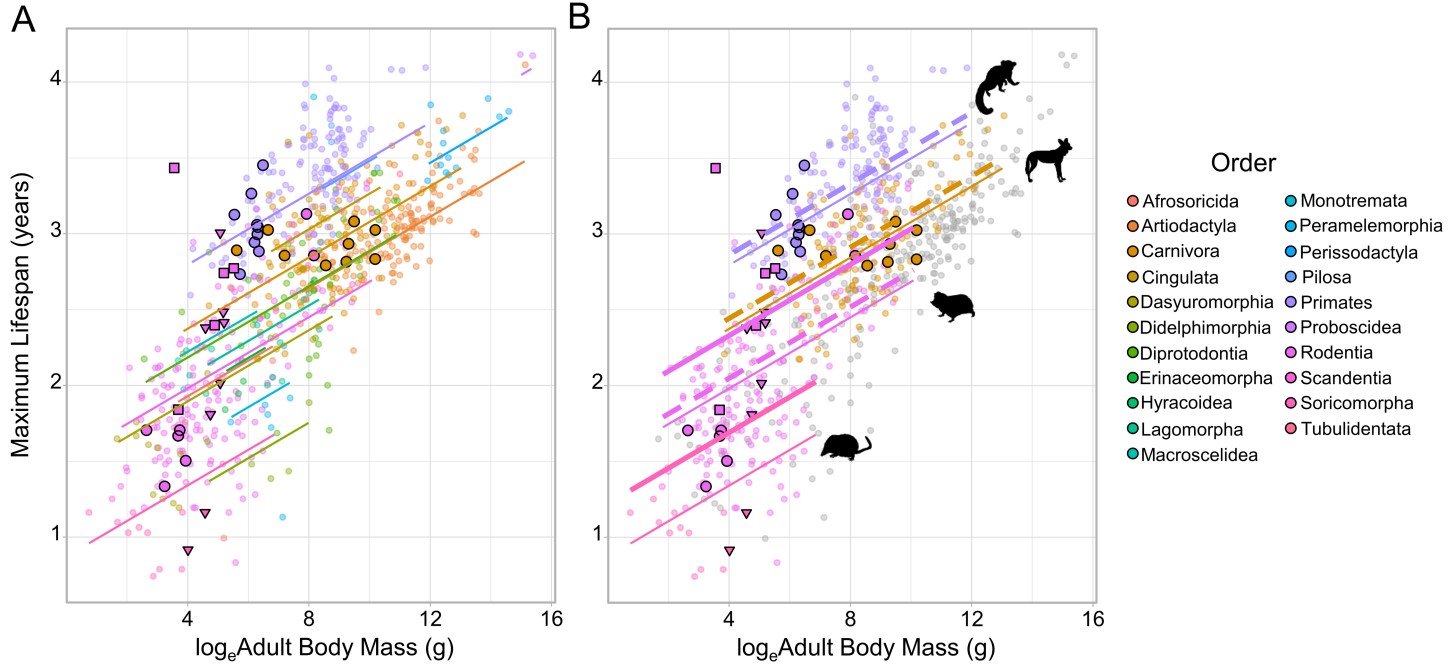

**Figure 1 The relationship between log$_e$ adult body mass and log$_e$ maximum lifespan for 719 terrestrial mammals.** (A) Species are coloured according to Order. Because of the strong phylogenetic contribution to maximum lifespan, the change in lifespan with increasing mass is predicted relative to the species with the median random effect estimate within each Order (as extracted from the MCMCglmm model fitted to the full dataset). Cooperative breeders and subterranean feeders have been given larger, highlighted points (cooperatively breeding only - bold circles; subterranean only - bold triangles; cooperatively breeding and subterranean - bold squares). (B) The predicted effect of cooperative breeding (dotted line) and/or subterranean feeding (thick solid line) is only illustrated in Orders containing species that display either trait. The animal icons were taken PhyloPic: http://phylopic.org. Predictions were generated at the medium sample size for a ground-dwelling mammal.

species with a 'small' sample size (10–100 specimens), 272 species with a medium sample size (100–1,000), and 123 species with a 'large sample size' (over 1,000 specimens). Of the 719 longevity records, 712 originated from captive populations (99.0%), four originated from wild populations, and three were of unknown origin. Sampling was not evenly distributed across cooperative and non-cooperative species: for cooperative species, large, medium and small lifespan sample sizes reflected 36.7%, 40.0% and 23.3% of species, as compared to 16.2%, 37.8% and 46.0% for non-cooperative species ($\chi^2_2$ = 10.38, $p$ = 0.006). All else being equal, longevity records for cooperative breeders are therefore likely to be higher simply due to sampling effort. Species that have been documented to breed cooperatively were defined as in the introduction ($n$ = 30 cooperative breeders). For each species further information was added on adult body mass, subterranean living and habitat. Subterranean living demarcated species as subterranean or non-subterranean feeders using information from several sources (*Begall, Burda & Schliech, 2007*; *Mittermeier, Rylands & Wilson, 2018*; $n$ = 13 subterranean species). For 'habitat', species were defined as arboreal ($n$ = 150), semi-arboreal ($n$ = 70) or ground-dwelling ($n$ = 499), using information from *Walker's Mammals of the World* (*Nowak, 1999*) and the *Handbook of the Mammals of the World* (*Mittermeier, Rylands & Wilson, 2018*).

## Global analysis

To investigate the influence of the chosen predictors on maximum lifespan, a global model was fitted which took the form:

$$\ln(\text{maximum lifespan}) \sim \ln(\text{adult body mass}) + \text{cooperative breeding} \\ + \text{subterranean} + \text{lifestyle} + \text{lifespan sample size}$$

Adult body mass was $z$-score transformed before model fitting. We implemented the model as a Bayesian phylogenetic mixed model (PLMM) using the *MCMCglmm* package (*Hadfield, 2010*) in R v3.6.2. PLMMs account for nonindependence between species by incorporating the phylogenetic tree as a random effect (*Hadfield & Nakagawa, 2010*). To account for phylogenetic uncertainty, we fitted the model on 100 randomly chosen mammalian trees from *Kuhn, Mooers & Thomas (2011)* and extracted the mean density of the combined posterior distribution using the *mulTree* Package (*Guillerme & Healy, 2014*). For each tree we ran two chains of 105,000 iterations with a burn-in of 5,000 and a thinning interval of 100 (generating a total 2,000 posterior samples for each parameter per tree). An inverse Wishart prior was chosen for the variance components ($V = 1$, nu = 0.02). All fitted models converged and the Gelman–Rubin statistic was always <1.1 for all parameters (*Gelman & Rubin, 1992*). The fixed effects were considered biologically significant when the 95% credible interval did not span zero. The marginal and condition $R^2$ for the model were calculated according to *Nakagawa & Schielzeth (2013)*.

To explore how model estimates were affected by the composition of the global dataset, we repeated the modelling approach on several subsets of the main dataset. We first repeated the analysis by excluding the naked mole-rat, or the Cape mole-rat. The naked mole-rat is a known outlier for mammalian longevity (*Buffenstein, 2005*), so a stronger case for a general effect of cooperative breeding on lifespan could be made if a cooperative breeding trend were present in the absence of this species. We also repeated the analysis without the Cape mole-rat to confirm that our inclusion of this species—with very low sample size informing the longevity record—did not unduly effect model estimates. We then re-ran the analyses to only include ground-dwelling species (i.e. excluded arboreal and semi-arboreal species to yield a dataset with $n = 499$ (with $n = 16$ cooperative breeders)), and also conducted a smaller analysis that only included five mammalian families that contain both cooperative and non-cooperatively breeding species (the Bathyergid mole-rats, the Callitrichid primates, the Canids, the Cricetid rodents, and the Herpestid mongooses; generating a dataset of 100 species, of which $n = 27$ were cooperative breeders). Because of the reduced dataset size for this latter analysis we only included fixed effects of cooperative breeding status, body mass and sample size.

## RESULTS

Across all the phylogenetic models fitted to the various datasets cooperative breeding had a positive effect on maximum lifespan (Table 1), but in all cases the effect size was small—reflecting at most a 3.1% increase in lifespan—and non-significant (Fig. 1; Table 1; Table S1). In contrast, subterranean feeding consistently explained significant increases

**Table 1 Phylogenetic analyses of maximum lifespan across terrestrial mammals.** Models were fitted using a phylogenetic linear mixed effects modelling framework, to a full dataset, a dataset excluding the naked mole-rat, and a dataset fitted to five cooperatively breeding families containing both cooperatively and non-cooperatively breeding species (Bathyergidae, Callitrichidae, Canidae, Cricetidae, Herpestidae). Estimates refers to the mean of the posterior distribution from models fitted to 100 different mammalian trees. Terms where the 95% credible intervals did not overlap zero were deemed biologically significant (highlighted in bold). The reference category refers to non-cooperatively breeding, ground-dwelling species.

| Model term | Global Dataset Mean estimate (95% CI) | Global dataset without naked mole-rat Mean estimate (95% CI) | Cooperatively breeding families Mean estimate (95% CI) |
|---|---|---|---|
| Intercept | 2.724 [2.123–3.324] | 2.727 [2.136–3.320] | 2.411 [2.032–2.790] |
| Adult body mass | **0.332 [0.275–0.391]** | **0.339 [0.282–0.370]** | 0.096 [−0.029–0.222] |
| Cooperative breeding status: cooperative | 0.065 [−0.200–0.149] | 0.056 [−0.029–0.141] | 0.061 [−0.033–0.156] |
| Fossoriality: subterranean | **0.348 [0.093–0.603]** | **0.269 [0.012–0.526]** | – |
| Lifestyle: semi-arboreal | 0.083 [−0.009–0.175] | 0.084 [−0.007–0.175] | – |
| Lifestyle: arboreal | **0.129 [0.024–0.235]** | **0.131 [0.026–0.235]** | – |
| Sample size: medium | **0.127 [0.089–0.166]** | **0.126 [0.088–0.165]** | **0.118 [0.027–0.204]** |
| Sample size: large | **0.207 [0.154–0.259]** | **0.202 [0.149–0.254]** | **0.287 [0.158–0.414]** |
| Residual variance | 0.015 [0.010–0.021] | 0.015 [0.010–0.021] | 0.015 [0.008–0.026] |
| Phylogenetic variance | 0.388 [0.314–0.464] | 0.378 [0.303–0.453] | 0.165 [0.096–0.274] |
| Marginal $R^2$ | 0.258 [0.190–0.336] | 0.270 [0.200–0.349] | 0.138 [0.044–0.290] |
| Conditional $R^2$ | 0.972 [0.958–0.982] | 0.971 [0.957–0.982] | 0.924 [0.852–0.969] |
| Species number | 719 | 718 | 100 |

in maximum lifespan across the candidate models (Fig. 1; Table 1; Table S1), and the influence of this term was maintained even after the removal of the known outlier species, the naked mole-rat. The effect of subterranean feeding ranged from 9.9% when the naked mole-rat was absent, to 12.8% when present. Beyond the effects of sociality and subterranean living, the models recovered a clear influence of sample size, such that species with more records contributing to their maximum lifespan had greater reported lifespans, all else being equal. The models also recovered the expected positive association between body size and maximum lifespan (with the exception of the cooperative families dataset, Table 1 and Table S1) and found that arboreal species were longer-lived than ground-dwelling species, with semi-arboreal species somewhere intermediate between these other two modes of life.

## DISCUSSION

The results presented in this study failed to find clear evidence for an association between cooperative breeding and increases in maximum lifespan across terrestrial mammals. Though the sample set of cooperatively breeding mammals is necessarily small, the effect size of any cooperative breeding effect on maximum lifespan is modest (at most 3.1%), and as such, it seems that the biological significance of any potential link between this aspect of mammalian sociality and lifespan remains obscure, particularly when viewed in light of ageing patterns in social insects (*Keller & Genoud, 1997*), or indeed birds (*Arnold & Owens, 1998*). In contrast, subterranean feeding was related to marked increases in maximum lifespan (12.8%), and this effect remained robust to the exclusion of the exceptional naked mole-rat. That subterranean species are longer-lived is in line with

evolutionary theories of senescence from which one can predict that lifestyles that reduce the level of extrinsic mortality—as is expected for species permanently inhabiting a subterranean niche—lead to the evolution of longer lifespans, and specific physiological adaptations in several long-lived subterranean species supports this view (*Kim et al., 2011*; *Fang et al., 2014*). Even so, there remains little quantitative information on the demography of subterranean taxa and this should be borne in mind when extrapolating our result more generally.

A consideration of the social organisation of cooperative societies can make sense of the reported null effect of cooperative breeding on lifespan. As helpers buffer the costs of reproduction in females (*Solomon & French, 1997*, *Russell et al., 2003*; *Creel, Mills & McNutt, 2004*), it could be argued that helpers might increase maximum lifespan by lightening the costs of reproduction for breeders, and recent work in Seychelles warblers *Acrocephalus sechellensis* has shown that the presence of a subordinate helper increases late-life survival in breeding females (*Hammers et al., 2019*). However, in cooperatively breeding mammals, the presence of helpers has also led to evolution of unusually high reproductive rates, and in some cases, atypically large litter sizes (*Barrette et al., 2012*; *Clutton-Brock et al., 2006*; *Clutton-Brock, 2016*). In addition, the extreme reproductive skew apparent in cooperative societies often engenders intense competition over mating opportunities which manifests in reproductive suppression, high levels of aggression, and the incidence of infanticide (*Clutton-Brock, 2016*). The presence of intense reproductive competition and high rates of reproduction are typically associated with high levels of energy expenditure and would classically be expected to accelerate rates of ageing and reduce lifespan, not prolong it (*Stearns, 1989*). Indeed, in wild Kalahari meerkats, females that experienced greater competition in early life displayed faster rates of reproductive senescence in later life (*Sharp & Clutton-Brock, 2011*). Previous work has also failed to find a link between group size and lifespan in mammals (*Kamilar, Bribiescas & Bradley, 2010*), so it is also not clear whether the large group sizes of cooperative breeders should necessarily contribute to their pace of life either.

The only family where cooperatively breeding species appear to live longer compared to non-cooperative breeders is in the Bathyergid mole-rats. Within this family the social taxa such as the naked mole-rat (31-year maximum lifespan) and the Damaraland mole-rat *Fukomys damarensis* (15.5 years), live markedly longer than the non-social, solitary members of the family such as the Silvery mole-rat *Heliophobius argentocinereus* (7.5 years), or the Cape mole-rat *Georychus capensis* (11.2 years). However, this comparison must be treated with caution, for unlike the social species, solitary mole-rats have attracted much less interest from researchers and are notoriously difficult to maintain in captivity, which will lead to large underestimates of the longevity of solitary species. There are nonetheless a number of illustrative cases outside of the African mole-rats that would tend to support a connection between subterranean living and lifespan extension. The subterranean and solitary plains pocket gopher (*Geomys bursarius*) and the Middle East blind mole rat (*Nannospalax ehrenbergi*) provide two examples, having been known to live for 12 years and 15 years, respectively (*Weigl, 2005*).

While our results are more suggestive of a role of subterranean living on lifespan, few subterranean mammals have been kept in captivity or been the focus of long-term individual-based studies. Moreover, a recent comparative analysis of reptiles failed to find a significant effect of fossoriality on lifespan (*Stark et al., 2018*), which the authors reasoned could be because any reductions in predation risk brought about through fossoriality are offset by high metabolic costs of burrowing. Taken together it therefore seems premature to place judgement on the role of subterranean living on ageing patterns in mammals until higher resolution data is collated from a larger number of species which permanently inhabit a subterranean niche, both in the wild and in captivity.

Overall, given current longevity data there is little evidence to support the hypothesis that cooperatively breeding mammals live longer than their non-cooperative counterparts. Whether this is true for other, more robust metrics of ageing will be an important test of the generality of this finding.

## ACKNOWLEDGEMENTS

JT would like to thank Prof. Tim Clutton-Brock, Louis Bliard and three reviewers for their comments on the manuscript.

### Funding

This work was funded by a Natural Environment Research Council Doctoral Training Programme (PFAG-043) awarded to Jack Thorley, who also received support from an ERC Advanced Grant No. 742808 awarded to Tim Clutton-Brock, University of Cambridge. The funders had no role in study design, data collection and analysis, decision to publish, or preparation of the manuscript.

### Grant Disclosures

The following grant information was disclosed by the authors:
Natural Environment Research Council Doctoral Training Program: PFAG-043.
Tim Clutton-Brock, University of Cambridge: 742808.

### Competing Interests

The author declares that they have no competing interests.

### Author Contributions

- Jack Thorley conceived and designed the experiments, analysed the data, prepared figures and/or tables, authored or reviewed drafts of the paper, and approved the final draft.

### Data Availability

Data is available in the Supplemental Files.

## Supplemental Information

Supplemental information for this article can be found online at http://dx.doi.org/10.7717/peerj.9214#supplemental-information.

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
