# Peer review of "The case for extended lifespan in cooperatively breeding mammals: a re-appraisal"

_PeerJ, doi:10.7717/peerj.9214_

## Round 0.1 · original submission · Major Revisions

Although this manuscript is an interesting and timely contribution, there are quite a few issues that the reviewers have raised that need to be addressed. One thing that will need to be adjusted a bit is the tone, and as one reviewers adds, and some clarity in other sections. However I also feel that the reviewers comments would significantly improve the content, and if you can pay special attention to remaining neutral in expression that would also be helpful.

Reviewer 1 ·

Basic reporting

The paper is well written and unambiguous, revealing that the author does not agree with previously published findings and that this paper, serves as a commentary on earlier published work.

Experimental design

This paper focuses upon an interesting idea raised by Austad, Healy and others that social /herd living animals have enhanced longevity relative to more solitary species. The author specifically focuses on those species that show cooperative breeding and uses as a data source the information accrued by de Magalhaes on the AnAge website. I have one major concern that I am sure the author can easily address, assuming he has not yet already done this. That is has he undertaken the appropriate power analyses to evaluate if he would be able to pick up a statistically significant difference between cooperative breeders and non-cooperative breeders if there indeed was one? With such limited data sets and relatively unreliable maximum lifespan estimates for some of the species (small sample sizes, not kept long enough in captivity to get an accurate estimate of max. lifespan) does he truly have enough power to categorically state there is no small, significant biologically meaningful difference between these two modes of behavior.
The authors also do not distinguish between eusocial and cooperative breeding.

Validity of the findings

If Dr Thorley has done the power analyses and determined the power needed to yield a significant change and met those criteria, I would be in strong support of publication. Without this analyses it is impossible to gauge the scientific merits of these findings.

Additional comments

Minor comments:-
The author cites Dammann's critique of the Ruby et al., paper without also citing the response to that commentary that refutes their claims by using additional analyses, creating a subjective bias in favor of the conclusions of this paper.
Data from solitary species, especially fossorial or strictly subterranean rodents in Walker's book and on the AnAge website differ greatly, reflecting the anecdotal nature of maximum lifespans with few substantiated records of truly maximum lifespan. No mention is made how these widely different data sets for often the same species are reconciled.

Reviewer 2 ·

Basic reporting

All good here

Experimental design

Conceptual issues aside (see general comments), here are some technical suggestions:
- Habitat categories are somewhat confounded with fossoriality. Arboreal or semi-arboreal species cannot be fossorial. Have you checked the VIFs for these models?
- You provide subsequent tests separating by family. What about providing an additional test where you only test this idea among terrestrial species and group them by fossorial or not? This seems like a better dataset for this question given the indication that fossoriality is an issue.
- MCMCglmm uses a Bayesian framework so it seems weird to refer to the p-values it provides in the same way you refer to those provided by the other (frequentist) methods. At the very least, it would be important to remind the readers in the manuscript what the p-value means in the MCMCglmm context. I would even suggest that model selection in that case would be better done using BIC or log-likelihood.
- On that regard… the stats you report in Table 1 suggest that you ran a fully parameterized model and simply reported its findings without attempts to reduce the models by removing non-significant terms – or using BIC-based model selection (please excuse me if I am missing something). If that is the case, then this could be problematic for your particular claim of a null result. For example, in the caper findings we see a marginal p-value for cooperative breeding (0.09) that could potentially cross the 0.05 threshold if non-informative parameters were to be removed from the model. Did you try that? The same could be said for the credible interval of coop bdg in the MCMCglmm output. It seems important to check that things don’t change after applying proper model selection algorithms
- It would also be nice to incorporate phylogenetic uncertainty into your analyses.

Validity of the findings

See general comments

Additional comments

In “Do cooperatively… live longer” the author critically re-evaluates the link between sociality and lifespan. There are several things to like about this paper, including the author’s care to redefine fossoriality as applicable only to species that spend significant amounts of time foraging underground (a more conservative definition than previous studies), the confirmation of findings with multiple statistical methods and the explicit correction for sample size in the estimate of lifespan. Other than a few minor technical comments provided below, I find the statistical approach sound. I am also quite aware that in this particular case, a null result is important and should be published. Having said that, there is one critical aspect of this study (and its predecessors) that casts a significant shadow on this entire discussion: estimating lifespan from data points collected in captivity simply does not seem to capture what the underlying hypothesis is all about. Conditions in captivity are incredibly different than in the wild: researchers are explicitly making active efforts to keep captive individuals alive, well-nourished and free from disease. At best, captive measures could be seen as a metric of lifespan potential, which is what the author has done, and differences between species could be interpreted as differences in how long a life cycle could be. In my opinion, though, the problem here is that the hypothesis being tested is likely to relate not to ultimate potential but to expressed one. As the title says, the idea being tested is not about how different species can live under idealized conditions but rather about whether social species actually do live longer in the wild. Sociality is thought to buffer the very things that captive conditions eliminate (disease, predation, food stress) so it is not very convincing to say that you don’t see any differences between social and non-social species when you take all of those issues away. This, to me, seems like a critical flaw even if, as the author explicitly mentions in the conclusion, you are only trying to point out that you think there is no evidence for a positive effect of cooperative breeding on lifespan in mammals.

·

Basic reporting

This paper is a re-appraisal of a previous analysis Healy 2015. However while this analysis does indeed reanalyse the question referred to in that analysis I believe it misrepresents both Williams and Shattuck’s and my comments and the appropriate context needs to be given here. See "General comments for the author" for more information.

Experimental design

See "General comments for the author" for other comments relating to category definitions.

Line 122-125: While I understand the retention of a the Cape mole-rat is due to the lack of information regarding African mole-rats, to show that the inclusion of this species does not affect the results a sensitivity analysis should be run with this species removed.

Line 135-137: How exactly was Subterranean defined. For example, is it defined differently to that stated in Healy et al 2014. Given that these definitions are the main difference to the previous analysis they need to be clear.

Lines 147: Was lifespan z-transformed, if not why not?

Line 158-162: Give how many chains were run and how convergence between chains was tested.

Lines 164-173: Why was the residual and t-test approach used instead of adding an additional cooperative breeding term in the general model. Also why was phylogeny not corrected, particular as the PLMM models show high phylogenetic signals.

Validity of the findings

All data have been provided, apart from some issue I have mentioned (convergence testing, number of chains, use of t-test) in Experimental design, the general analysis are statistically sound. See below regarding to how the conclusions are stated and linked to the litrature.

Additional comments

When I was asked to review this paper I was very happy to see that someone had reanalysed a comparative analysis that formed part of a comment and response to an article I published on maximum lifespan in 2014. In particular, the comment (Williams and Shattuck 2015) and response (Healy 2015) were both relatively limited analysis that mainly served to raise the issue that sociality may be a confounding factor for increased lifespan in fossorial mammals. While this analysis does indeed reanalyse the question using a different categorisation of sociality, something I greatly welcome, unfortunately I believe it misrepresents both Williams and Shattuck’s and my comments and I this needs to be put into its proper context and given a tone that appropriately reflects this context.

To give the full context the analysis this paper sets out to conduct a re-appraisal of is my 2015 analysis (Eusociality but not fossoriality drives longevity in small mammals). My 2015 analysis was published as a reply to Williams and Shattuck’s comment and analysis (Ecology, longevity and naked mole-rats: confounding effects of sociality?) which in turn was a comment in relation to the original manuscript Healy et al. 2014 (Ecology and mode-of-life explain lifespan variation in birds and mammals). In this string of analysis Williams and Shattuck highlighted the potentially confounding effect of sociality and showed it had a positive effect on lifespan using a new dataset derived from, but with several important changes, the original dataset in Healy et al. 2014. These changes consisted of using different categorisations for fossoriality and only focused on small mammals (< 60kg) with the potential for fossoriality. My 2015 analysis simply reanalysed the data using a MCMCglmm approach which also incorporated error associated with phylogenetic relationships. My analysis found that fossoriality no longer had a significant effect on lifespan while sociality did.

This manuscript here needs to clearly highlight this context in several ways. For example, while the origin of this analysis stems from Williams and Shattuck 2015, this work is only mentioned in passing in the last paragraph of the discussion (the reference is also missing from the reference list). This paper needs to refer to Williams and Shattuck 2015 throughout and be clear with regards to that being the initial and main analysis with regards to the Healy et al 2014 string of analysis associated with sociality. I think it would also greatly benefit this manuscript to focus on how the analysis here builds on, but differs, from their analysis, and to a lesser degree to Healy 2015 (which only differed to Williams and Shattuck 2015 by using a MCMCglmm approach). Furthermore, the tone of this manuscript should more clearly recognise that these comments were never intended as a final say or anything more than the start of a conversation regarding the role of sociality in comparative analysis of lifespan. In fact, the final paragraph of Healy 2015 highlights this “Williams and Shattuck’s work helpfully extends our analysis to include eusociality; however, the lack of correlation between fossoriality and longevity in this dataset requires further attention given the predicted reduction of extrinsic mortality associated with burrowing. Analysis of fossorial species in other groups such as reptiles, along with the inclusion of more detailed life-history data, may yield more insights into the role of fossoriality in life-history evolution.”

Overall, I think the tone of this manuscript of setting Healy 2015 up as a stand alone analysis to be taken apart misses on the opportunity to build a more constructive paper highlighting the clear gaps in both those previous analysis and to point a more constructive way forward for the field. In particular, I think this paper would be greatly improved if it focused on a message such as “Here we use a more stringent definition of sociality along with a definition of fossoriality more closely related to the original fossoriality in Healy et al 2014 to test the role of cooperative breeding and fossoriality in driving longevity. We also extend the analysis to mammalian groups with no fossorial species to test whether cooperative breeding is associated with increased lifespan across mammals.” By framing it this way the focus would be squarely on the work presented here and not on the limitations of previous short comments and replies.


I give more detailed comments below.

Lines 86-91: mention the more relaxed definition of fossorial used in the 2015 analysis. I refer the author to Williams and Shattuck 2015 who combined the original fossorial and semi-fossorial categories from the Healy et al 2014 into one category. I agree that this is too relaxed a definition and a more restricted term, as used here, is more appropriate for the question. However, Williams and Shattuck 2015 need to be cited here, and the fact that a fossorial category, similar to the one used here, was initially used in the 2014 manuscript.

Lines 95-100: It should be mentioned that, while sample size was not included as a factor, it was considered in the methods of Williams and Shattuck 2015 and Healy et al 2014, with Healy 2015 using the data from Williams and Shattuck 2015. As it stands it reads as if no consideration was made for sample size or data quality in previous analysis.


Lines 100-103: Again out of context this line misses the reasoning from Williams and Shattuck 2015 for not including Callitrichid primates. I refer the author to thier paper which mentions “Since fossorial and eusocial-like species are restricted to smaller body sizes, we do not include mammals larger than 60 kg, a boundary that is approached by the largest semi-fossorial and eusocial-like mammals, aardvarks and humans, respectively.”


Line 195-198: Healy (2015), following from the data from Williams and Shattuck 2015, does not incorporate foraging mode directly as the analysis was conducted on a sub group effectively excluding arboreal species. Hence, to suggest that these analysis failed to control for variation in foraging mode is misleading. I also don’t understand the comment regarding the failing to consider the leveraging effect of exceptionally long lived specie, such as the naked mole rat, given these species were central to the discussion and analysis of Healy et al 2014, Williams and Shattuck 2015 and Healy 2015. Furthermore, my 2015 paper specifically tests the effects of incorporating uncertainty from phylogenetic relationships, to help account for such “outlier groups”. Also it is not clear how this outlier leverage effect was controlled for in a way that was not controlled for previously. I would suggest removing this line.

Line 211-219: It should be mentioned that the definition of cooperative breeding here may be more relaxed than Williams and Shattuck 2015. It would be useful to compare the species defined as eusocial in Williams and Shattuck 2015 to the species defined as cooperative breeding here to see if there is a difference as this might also explain the different results.

Line 221-232: The author suggests here that the previous analysis were in affect pre-mature. This again ignores the context of both comments and also the fact that both papers highlight that more work needs to be done, again referring to my 20015 comment I end on the note “Analysis of fossorial species in other groups such as reptiles, along with the inclusion of more detailed life-history data, may yield more insights into the role of fossoriality in life-history evolution “. I suggest a more constructive tone should be used in this paragraph pointing towards how the field may advance. For example, it could refer to recent relationship found between fossoriality and life span in reptiles (Stark et al 2018). Unnecessary tone of what the author determines as useful or not useful are not helpful here in my opinion. For example, the comments “This being the case, it not it is not particularly useful, for example, to claim that placental moles are on average characteristically short-lived in the context of all mammals (as per Williams & Shattuck 2015). For one, moles are rarely studied, and a cursory glance at another ageing database (Jones et al. 2009) indicates that one of the few well-studied species, the European mole Talpa europaea (not present in this study), can live for seven years, a relatively long time for an 80g insectivore.”. For one, placental moles are indeed on average characteristically short-lived in the context of all mammals. Using your example an 80g mammal living for 7 years would still have a negative residual based on the Healy et al 2014 analysis. Hence, while 7 years might be relatively long for an 80g insectivore it is not long in the context of all mammals. This statement doesn’t so much bother me in its inaccuracies but more so in its tone and quite frankly aggressive tone to another author when it is in no way merited. I think it would be far more helpful to make the useful and important point regarding the difficulties and pitfalls of using maximum lifespan measured for species that have not received much research attention.


Lines 235-236: The author suggest there is little evidence that cooperatively breeding mammals live longer despite the fact that their own models found a significant (and may still be so in the Bayesian model if the chains never converged), albeit small, effect for this. This is clear cheering picking of the results, especially given the context of results in other papers. It would be more consistent to say that the effects of cooperative breeding on longevity are inconsistent across models here and in all cases there is only small effect sizes related to them.

Lines 237-239: “Future studies assessing the role of sociality or mode of life on lifespan variation should consider the availability and quality of data before conducting large-scale comparative analyses.” While this is true, and normally I would not think anything about this statement, given the tone of this manuscript this reads as if previous studies have ignored this. Given the wealth of comparative analysis which strongly point the dangers and pitfalls of using maximum longevity, such as from Anage itself, I think a rephrasing this to “Echoing previous studies (de Magalhães et al 2007), future studies assessing the role ….”


Kevin Healy

---

## Round 0.2 · Minor Revisions

Unfortunately your previous academic editor is not currently available and so I have taken over the handling of your article and review. I have familiarized myself with all the materials associated with your submission including your original submission and the three associated reviews and your revised resubmission. Furthermore, two of the three reviewers who reviewed your original submission have provided feedback on your revision and I have carefully read their most recent feedback.

Both reviewers praised the work you had done on clarifying your article and toning down your messaging. I agree with them that the article is clearly stated and the methods are well presented. However, both reviewers also have a number of outstanding comments that I request you address. Most notably is a call for you to include consideration of the source of the data in your analyses i.e. whether the data pertain to wild or captive animals. I agree that this would be valuable.

Furthermore, both reviewers highlighted that there were a number of typos and grammatical errors within your article. I also found some. For example on line 213 you state "(Table 1, Table 1)". I presume the latter should be Table S1, but please clarify this and proof read your entire article with care. Finally, I suggest that given the content provided in Fig. 1, you should also reference it in your results as well as in your methods section.

I look forward to receiving your revision.

Reviewer 1 ·

Basic reporting

For the most part OK, a few typos/grammatical errors that should be picked up in final revision

Experimental design

All good

Validity of the findings

With caveats that the maximum lifespan data set may not be as accurate given that these are captive records and AnAge has not necessarily updated as things have changed. It is a vast improvement on Walker's

Additional comments

The authors have addressed some of my comments or at least provided a convincing rebuttal.

The track changes version is not identical to the PDF! For example line 55 includes the Damman and Ruby response citations, while in the track changes version it does not and is far less confusing to read. It seems very strange to have a set of references cited for a finding, in this case mortality rate followed by a “but see”. If you want to make a comment about these data write a sentence to state your position, as it stands this is most confusing to the reader.

There are also several grammatical errors or typos. For example Line 157 “In general information The only exception to this data restriction”; eLine 210. .. “had a positive effect on maximum lifespan that was positive”.. the sentence should end at maximum lifespan.

It is my understanding that Damman et al has shown that cooperative breeder Fukomys anselli live 20 years as do the solitary spalax, so in many cases of the small subterranean cohort, lifespans are likely to be underestimates. This should be strongly emphasized but does not detract from the analyses undertaken using the not so up to date AnAge data set.

·

Basic reporting

Overall the author has toned down the article and placed the previous analysis into a better context, however there are still a few issues with regards to this that I highlight below. The focus on using just MCMCglmm does simplify the analysis and makes the results more easily interpretable.

I outline my comments in more detail below.

Experimental design

Overall, I think that these analysis are generally sound. However I do think given the context of this paper that the issue of captivity as highlighted by another review should be handled better. I would also like to point out the power analysis suggested is possible (a power analysis is simply simulation of how many more such species would need to exist and does not require more species to actually exist). However I do agree it is not a trivial problem to fix for phylogenetic comparative model (although still possible).

Validity of the findings

As the main goal of this paper is to re-analysis previous questions and updating their shortcomings regarding data quality and the definition of terms, I think it is important to address the comments of reviewer two explicitly in the analysis. The comment outlines that comparing cooperative breeders and non-cooperative breeders from captivity data may have an addition problem due to the usual effects of using captivity data for maximum longevity. In effect, we expect that cooperative breeders would not greatly increase maximum longevity in captivity when compared to non-cooperative breeders. If this was the case and the majority of the data was from captive sources any increase in maximum lifespan associated with cooperative breeders could easily be masked in the analysis. I suggest including captive versus non-captive as an additional explanatory factor in the analysis with an interaction term with cooperative breeding. If this really is an issue you would expect a negative interaction associated with non-captive measures of longevity for non-cooperative breeders.

Additional comments

In Figure 1 there are OLS lines fitted for each order, yet the analysis is a MCMCglmm. I imagine this is a hangover from the previous iteration of the paper were OLS models were used. I would recommend removing these fitted OLS lines and only used estimated from the main MCMCglmm model. It would also be far more informative for the reader if both cooperative and fossorial species were marked out in the graph and the overall mass longevity line fits for both cooperative breeding and fossoriality were included.

With regards to the response from the author“In my opinion not enough was done to control for sample size, as I try to make clear in the introduction (line 137).” in relation to my comment that sample size was considered in the previous papers. I would like to point out that in Healy et al 2014 two analysis were run; one using max longevity for species with only 10 records (small) and a second analysis that only included species with 100 records (large). This effectively boils down to include an explanatory factor of sample size with two levels (even if they are not explicitly included in the model we effectively manually partitioning variance in the data). In this analysis, while an explanatory variable for sample size is included, it only has three levels, so to suggest, as the author does, that this analysis has fixed the problem of sample size in a way that others have not is over egging the contribution in my opinion. It would be comparable to someone re-running the authors analysis with sample size included as a continuous variable and suggesting that “although previous studies tried to overcome possible issues of sample size” “not enough was done to control for sample size”. I would suggest to tone down the line 214-224 to something along the lines of “included sample size in their analysis, using sensitivity analysis relating to sample size, here we explicitly include sample size in each model.”


Minor comments.
Line 247: remove “In general information”
Line 380 “Healey” should read as “Healy”
Line 381-382: I assume there is a mistake in the numbers for the chains as 105000 iterations with a burnin of 5000 and thining of 100 would only generate 1000 posterior samples for each parameter and not 200000
Line 383: Gelman-Rubin statistic needs citation.
Line 552: For clarity include “compared to non-cooperative breeders” in the line “The only family where cooperatively breeding species appear to live longer is in the bathyergid mole-rats.”

---

## Round 0.3 · accepted · Accept

Thank you very much for your careful responses to each of mine and the reviewers' feedback. It is my pleasure to accept your article for publication in PeerJ.

However, I note that where you now describe the origin of the data in your data set (thank you for including this!) there is a typo. Your statement reads "Of these 719 longevity records, 712 originated from captive populations (99.0%), 3 originated from wild populations, and 3 were of unknown origin." However, I believe, from your note to me in your rebuttal document, that there were 4 records from wild populations. Please correct this prior to publication, thank you.